# Experiences of people living with HIV who have participated in psychological interventions: Protocol for a qualitative meta-analysis

Cristian Ortega[1,2,3¤a], Jaime Garcia-Iglesias[3¤b]*, Felipe Concha[1,2], Francisca Mena[1,2‡], Alemka Tomicic[1,2‡]

1 Centre of Research in Clinical Psychology and Psychotherapy (CEPPS), University Diego Portales, Santiago, Chile, 2 Millennium Institute for Research on Depression and Personality (MIDAP), Santiago, Chile, 3 Centre for Biomedicine, Self and Society, Usher Institute, College of Medicine and Veterinary Medicine, University of Edinburgh, Edinburgh, Scotland

☯ These authors contributed equally to this work.
‡ These authors also contributed equally to this work.
¤a Current Address: Faculty of Psychology, University Diego Portales, Santiago, Chile.
¤b Current Address: Usher Institute, College of Medicine and Veterinary Medicine, University of Edinburgh, Edinburgh, Scotland.
* jgarcia6@ed.ac.uk

## Abstract

### Introduction

Mental health problems among people living with HIV have been widely documented, including a high burden of common mental disorders. Despite the demonstrated effectiveness of various psychological interventions, there is limited understanding of how these interventions are experienced and evaluated by participants themselves. A key factor in ensuring successful implementation lies in how individuals interpret and emotionally respond to an intervention—specifically, whether they perceive it as meaningful, appropriate, and aligned with their expectations and needs. This qualitative meta-analysis aims to explore the subjective therapeutic experiences of people living with HIV who have participated in psychological interventions, with a particular focus on their acceptability and the perceived impact on psychological well-being.

### Methods and analysis

A qualitative meta-analysis will be conducted following the descriptive-interpretive approach, which enables the identification and synthesis of meaning units from participants' narratives. The review will include qualitative and mixed-methods studies, with a clearly distinguishable qualitative component, published between January 1, 2000 and February 23, 2026, focusing on the therapeutic experiences of people living with HIV who have received psychological interventions for common mental health problems. A comprehensive search will be performed in MEDLINE (via PubMed), MEDLINE (EBSCOhost), Scopus, Web of Science, APA PsycINFO and SciELO.

**Data availability statement:** No datasets were generated or analyzed during the study as this is a literature review protocol. All relevant information is within the paper and its Supporting Information file.

**Funding:** This Qualitative Meta-Analysis is funded by University Diego Portales, Centre of Research in Clinical Psychology and Psychotherapy (CEPPS) and ANID – Millennium Science Initiative Program / Millennium Institute for Research on Depression and Personality - MIDAP AIM23_0002 The funders had no role in study design, data collection and analysis, decision to publish, or preparation of the manuscript.

**Competing interests:** The authors have declared that no competing interests exist.

Methodological quality will be assessed using the CASP tool, and confidence in the findings will be evaluated through GRADE-CERQual. The data analysis will follow the descriptive-interpretive meta-analysis approach. Data management and thematic analysis will be supported by Microsoft Excel, Rayyan and/or ATLAS.ti.

## Expected results

This review is expected to identify key experiential themes that reflect how people living with HIV perceive and evaluate psychological interventions, including the aspects they find most helpful, challenging, or relevant. Findings will contribute to a better understanding of the acceptability of such interventions and offer practical insights for improving their design, delivery, and contextual adaptation.

## Dissemination

The results of this review will be disseminated through publication in a peer-reviewed journal and presentations at scientific conferences related to mental health, HIV, and qualitative research. The findings are expected to inform the development of more acceptable and culturally sensitive psychological interventions for people living with HIV.

---

## Introduction

Mental health problems among people living with HIV have been widely and systematically reported globally [1,2], with a high burden of common mental disorders and related difficulties in this population [3]. These include depressive and anxiety disorders [4,5], substance use disorders [6], and suicidal ideation [7]. Poorer mental health has been associated with challenges in HIV care and disease management. For example, a study conducted in China [8] reported that depressive symptoms among people living with HIV were associated with poorer antiretroviral therapy (ART) adherence, slower viral suppression, faster progression to AIDS, and higher mortality risk.

Among the psychological mechanisms implicated in common mental health problems among people living with HIV, HIV-related self-stigma has been identified as a key factor. Self-stigma is a socially constructed determinant of mental health and refers to the internalization of negative beliefs, thoughts, and behaviours associated with HIV. It may manifest as shame, guilt, feelings of contamination, self-contempt, low self-esteem, and self-rejection [9]. Higher self-stigma has been linked to poorer health-related quality of life, including more negative illness perceptions and dissatisfaction with sexual and romantic life [10–12], as well as reduced adherence to pharmacological treatment [13,14]. Overall, self-stigma may adversely affect both mental and physical health outcomes across diverse sociodemographic groups.

Psychological interventions have shown evidence of effectiveness in addressing mental health problems among people living with HIV, using various therapeutic approaches (e.g., time-limited dynamic psychotherapy [15]; cognitive-behavioral

therapy [16]; interpersonal psychotherapy [17]). These interventions have been delivered individually [18] and in groups [19], face-to-face [20] and remotely [21], across urban [22] and rural contexts [23], and among diverse populations (e.g., women, [24] Knettel et al., 2020; gay, bisexual, and other men who have sex with men, [25] Barrington et al., 2023; transgender individuals, [26] Iyer et al., 2024).

In light of the above, it has become increasingly important to adapt psychological interventions that are grounded in empirical evidence, ensuring both their feasibility and acceptability among target populations. For instance, Zhao et al. [27], through a systematic review and meta-analysis, concluded that mind-body therapy, interpersonal psychotherapy, cognitive behavioral therapy, supportive therapy, and educational interventions are effective non-pharmacological approaches for the treatment of depression in this population. However, in terms of acceptability, defined as the extent to which people delivering or receiving a healthcare intervention consider it to be appropriate based on their anticipated or experienced cognitive and emotional responses [28], both supportive therapy and interpersonal psychotherapy were found to be significantly less acceptable than most control conditions. This finding suggests that although certain interventions demonstrate clinical efficacy, they may not be well received by participants, potentially resulting in reduced adherence, discomfort, or negative perceptions of the therapeutic process. Consequently, it is crucial to further explore the conditions under which psychological interventions are deemed acceptable, particularly from the perspective of individuals who experience internalized stigma.

Despite the availability of evidence-based treatments and robust evidence of their effectiveness in managing mental health disorders through objective outcome measures, there remains a limited understanding of the lived experiences of people living with HIV who have participated in psychological interventions. Gaining insight into these experiences is essential to improving the acceptability of such interventions.

Accordingly, the present qualitative meta-analysis aims to systematically identify, synthesize, and critically appraise qualitative evidence on the acceptability of psychological interventions for common mental health problems in people living with HIV. Specifically, it seeks to explore how these interventions are perceived, which elements are considered beneficial or challenging, and how they contribute to psychological well-being, based on participants' accounts.

To address these gaps, this review will follow the qualitative meta-analysis approach proposed by Timulak and Creaner [29], which emphasizes the systematic identification, categorization, and synthesis of meaning units derived from participants' narratives. This method is particularly appropriate for exploring subjective experiences, as it allows for the construction of thematic categories grounded in participants expressed perspectives. By applying this meaning-oriented synthesis, the review aims to generate nuanced insights into how people living with HIV experience psychological interventions, and what factors influence their acceptability.

### Research objectives

**Primary research question.** What are the lived therapeutic experiences of people living with HIV who have participated in psychological interventions, as described in qualitative studies?

**Primary objective.** To conduct a qualitative meta-analysis of published research that explores the lived experiences of people living with HIV who have engaged in psychological interventions targeting common mental health problems.

**Secondary objectives:**

• To identify the elements of psychological interventions that participants perceive as beneficial or challenging.

• To understand how these elements influence participants' psychological well-being, from their own perspectives.

### Materials and methods

This qualitative meta-analysis protocol was registered in the International Prospective Register of Systematic Reviews (PROSPERO) on June 2, 2025, under the registration number CRD420251065815.

This protocol was developed in accordance with the Preferred Reporting Items for Systematic Reviews and Meta-Analyses Protocols (PRISMA-P) 2015 statement [30], which provides standardized guidance for preparing and reporting systematic review protocols.

## Eligibility criteria

To ensure methodological rigor and alignment with the qualitative nature of this review, eligibility criteria will be guided by the SPIDER tool (Sample, Phenomenon of Interest, Design, Evaluation, Research type), which is considered appropriate for identifying relevant qualitative studies [31]. This framework supports the inclusion of studies that emphasize participants' subjective experiences and meanings—consistent with the objectives of this qualitative meta-analysis and the analytic approach proposed by Timulak and Creaner [29].

**Sample (S).** Studies must include adult participants (aged 18 years or older) living with HIV who have received a psychological intervention. There will be no restrictions based on gender, geographic location, cultural background, or risk group (e.g., men who have sex with men, transgender individuals, women, people who use drugs). Studies focusing on key populations particularly affected by HIV will be included, provided they meet the other criteria.

**Phenomenon of Interest (PI).** The central phenomenon of interest is the participants' lived experiences with psychological interventions aimed at addressing common mental health problems—such as depression, anxiety, substance use, or suicidal ideation. The review focuses on how individuals perceive, describe, and make sense of these interventions in the context of their broader experience of living with HIV, including factors related to self-stigma and mental well-being.

**Design (D).** Eligible studies must employ a qualitative research design, including but not limited to phenomenology, narrative inquiry, grounded theory, ethnography, or qualitative content analysis. Mixed-methods studies will be included only if they report qualitative data separately and provide sufficient detail to allow for independent extraction and synthesis.

**Evaluation (E).** Studies must report participants' subjective evaluations of the intervention, including perceived acceptability, satisfaction, therapeutic value, challenges, barriers to engagement, emotional responses, or perceived changes in psychological well-being. Studies that only assess intervention effectiveness using quantitative outcomes (e.g., symptom scores) without qualitative insight into participant experience will be excluded.

**Research type (R).** This review will include primary qualitative research studies and mixed-methods studies with a clearly defined and analyzable qualitative component. Editorials, theoretical papers, conference abstracts, and purely quantitative studies will be excluded.

## Information sources and search strategy

A comprehensive literature search will be conducted using the following electronic databases: MEDLINE (via PubMed), MEDLINE Complete [EBSCOhost]), Web of Science, Scopus, APA PsycINFO, and SciELO. These databases were selected to ensure broad coverage of biomedical, psychological, and social science research relevant to the topic. The search will include peer-reviewed journal articles published in any country and in any language. In addition, the reference lists of all included studies and relevant systematic reviews will be screened for potentially eligible articles. Grey literature (e.g., dissertations, theses, unpublished reports) will not be included, as this review is limited to peer-reviewed qualitative and mixed-methods studies to ensure methodological rigor and reporting quality. The full search strategy developed for MEDLINE Complete (EBSCOhost) is provided in S1 Table. This strategy will be adapted, as appropriate, for the other databases included in the review.

Recognition of the mental health needs of people living with HIV has been increasingly institutionalized since the early 2000s. Both the World Health Organization and UNAIDS have emphasized that mental health is a critical component of comprehensive HIV care, as outlined in key strategic and technical documents [32,33]. Therefore, this review will include studies published between January 1, 2000 and February 23, 2026. This temporal restriction ensures the inclusion of

research conducted within contemporary treatment contexts, where the psychological experiences of people living with HIV are shaped by long-term management rather than acute or terminal care settings.

## Data management/selection process

The results of the literature search will be imported into Rayyan, a web-based citation management tool designed to facilitate systematic reviews. Duplicate records will be identified and removed prior to screening.

Title and abstract screening will be conducted by two reviewers. The lead reviewer will screen all retrieved records, while two additional reviewers will each independently screen 50% of the records. In this way, all records will be assessed by at least two reviewers to ensure reliability and reduce selection bias. Discrepancies will be resolved through group consensus. If consensus cannot be reached, the record will be retained for full-text screening.

Full-text versions of all potentially eligible studies will be retrieved and independently assessed for inclusion by two reviewers. In cases of disagreement, a third reviewer will be consulted to reach consensus. Reasons for exclusion at the full-text stage will be documented in detail.

The entire selection process will be presented using the PRISMA 2020 flow diagram [34] ensuring transparency and reproducibility.

## Data extraction/ data collection

Data will be extracted using a structured form developed in Microsoft Excel. To ensure consistency and reliability, two reviewers will independently extract data from a pilot sample comprising 10% of the included studies. A third reviewer will moderate discrepancies and finalize the extraction framework.

Given the anticipated heterogeneity across studies—in terms of design, populations, intervention types, outcomes, and qualitative data formats—the extraction process will be piloted and iteratively refined as needed during full-text screening.

Once the form has been validated, one reviewer will extract data from the remaining studies. A second reviewer will verify a random subset (at least 20%) to ensure accuracy. Any disagreements will be resolved through discussion or by involving a third reviewer.

## Data items

The following data items will be extracted from each included study, where available:

**General study characteristics.** Author(s), full citation, study title, journal, publication year, country and study setting, study period, funding sources, declaration of competing interests, study design or methodology, duration, and number and timing of follow-ups.

**Participant characteristics.** Description of the study population, including subpopulation or risk group (e.g., MSM, transgender women, women living with HIV), sample size (at baseline, post-intervention, and follow-up), age range, gender, ethnicity, method of recruitment, inclusion/exclusion criteria (especially those related to mental health conditions such as depression, anxiety, substance use, or suicidal ideation), and comorbidities.

**Intervention characteristics.** Description of the psychological intervention including its theoretical framework or orientation (e.g., CBT, interpersonal psychotherapy), delivery format (individual or group), setting (in-person, remote, or blended), platform/tools used (e.g., websites, SMS, mobile apps), number and duration of sessions, providers/facilitators involved.

**Evaluation and outcome information.** Data related to participant-reported experiences and evaluations of the intervention (or elements thereof), including perceived acceptability, usefulness, feasibility, accessibility, emotional or cognitive responses, facilitators and barriers, drop-out rates, and usability feedback. Where reported, relevant outcome measures (e.g., symptom reduction or service user satisfaction) will also be recorded, although these will be interpreted qualitatively rather than quantitatively.

## Outcomes and prioritisation

The primary outcome of this qualitative meta-analysis is the perceived acceptability of psychological interventions targeting common mental health problems (e.g., depression, anxiety, substance use, and suicidal ideation) among people living with HIV. Acceptability will be defined according to the framework proposed by Sekhon et al. [28] as the extent to which individuals delivering or receiving a healthcare intervention consider it appropriate, based on anticipated or experienced cognitive and emotional responses. This may be reported through participants' own reflections on emotional engagement, satisfaction, perceived usefulness, burden, intervention coherence, ethical alignment, and self-efficacy.

Secondary outcomes will include:

a) Participants' evaluations of the psychological impact of the intervention on their mental health and overall well-being, including experiences of change or stability in symptoms, coping skills, or stigma;

b) Experiences related to the accessibility, feasibility, and contextual relevance of the intervention, such as barriers to engagement, facilitators, or preferences for format and delivery mode (e.g., remote vs. in-person);

c) Interpersonal or relational outcomes, such as perceived support, therapeutic alliance, or group cohesion;

d) Broader social or structural experiences that influence engagement with psychological interventions, including stigma, discrimination, or intersectional vulnerabilities (e.g., gender identity, substance use, socioeconomic status).

## Risk of bias in individual studies

To assess the risk of bias in individual studies, the Critical Appraisal Skills Programme (CASP) checklist for qualitative research [35] will be used. This tool evaluates key methodological domains, including clarity of research aims, appropriateness of the design, recruitment strategy, data collection methods, consideration of the researcher-participant relationship, ethical issues, data analysis rigor, and transparency in reporting findings.

Two reviewers will independently appraise each included study using the CASP tool. Disagreements will be discussed and resolved through consensus, with a third reviewer consulted if necessary. While no study will be excluded based solely on its appraisal score, the results of the critical appraisal will be used to contextualize the findings of the synthesis and to assess the credibility, transferability, and dependability of the evidence.

A summary of the CASP appraisal ratings will be provided in a supplementary table for transparency.

## Data analysis and synthesis methods

The data analysis will follow the descriptive-interpretive meta-analysis approach proposed by Timulak and Creaner [29], which emphasizes the identification, organization, and synthesis of meaning units derived from participants' narratives. This approach is particularly well-suited for capturing the experiential and contextual richness of qualitative data while preserving participants' voices.

The meta-analysis will be conducted through the following steps:

1. **Familiarization with the data**

All qualitative findings (i.e., results sections, themes, and direct quotations) will be read thoroughly to achieve immersion in the data and contextual understanding of each study.

2. **Identification of meaning units**

Discrete segments of text (meaning units) that reflect significant aspects of participants' experiences with psychological interventions will be extracted. These units may include participant quotes or author interpretations and will be treated as the basic elements of synthesis.

3. **Preliminary categorization**

Meaning units will be grouped into lower-order categories based on thematic similarity. These categories will remain close to the participants' original expressions and language.

4. **Construction of core categories and interpretive themes**

Through iterative comparison and abstraction, core categories will be developed by grouping lower-order categories that reflect broader patterns across studies. Interpretive themes will be defined and refined collaboratively by the review team.

5. **Cross-case analysis and interpretation**

Themes will be compared across studies to explore contextual patterns (e.g., intervention type, population subgroup, delivery mode) and to identify similarities, divergences, or emerging subthemes. When possible, subgroup-specific patterns (e.g., by gender identity, sexual orientation, or substance use) will be explored to identify contextual differences in the acceptability and perceived impact of psychological interventions.

The meta-analysis will be managed and organized using Microsoft Excel and/or ATLAS.ti software, depending on the complexity of the dataset. Two reviewers will collaborate on the analysis process to ensure rigor and reflexivity. Discrepancies in interpretation will be resolved through discussion or adjudicated by a third reviewer if necessary. The goal of the qualitative meta-analysis is to generate an in-depth understanding of the acceptability and experiential impact of psychological interventions among people living with HIV, contributing to more context-sensitive and person-centered intervention design.

## Meta-bias

Potential qualitative meta-analysis bias will be mitigated through:

a) Comprehensive search strategy, including multiple databases and backward citation tracking to reduce the risk of missing relevant studies.

b) The inclusion of studies regardless of quality appraisal score, while using CASP results to contextualize the weight and credibility of findings.

c) Transparent reporting of the meta-analytic process, including how meaning units and themes were developed, how disagreements were resolved, and how researchers' interpretations were checked for reflexivity.

## Confidence in cumulative estimate

The confidence in the findings of this qualitative evidence will be assessed using the GRADE-CERQual (Confidence in the Evidence from Reviews of Qualitative Research) approach, which provides a transparent and systematic method for evaluating the trustworthiness of individual review findings [36].

CERQual assesses four components for each synthesized finding:

1. **Methodological limitations**

The extent to which problems in the design or conduct of the primary studies may decrease confidence in the finding. This will be informed by the results of the CASP appraisal.

2. **Coherence**

The degree to which the data supporting the review finding are consistent and well-explained across multiple studies.

3. **Adequacy of data**

The richness and quantity of data supporting each review finding, including the number of studies, participants, and contexts represented.

4. **Relevance**

The extent to which the evidence supporting a review finding is applicable to the context specified in the review question (e.g., people living with HIV who have received psychological interventions for common mental health problems). Each synthesized finding will be assigned an overall level of confidence: high, moderate, low, or very low. These ratings will be justified and summarized in a CERQual Summary of Qualitative Findings (SoQF) table, following the recommended template by the GRADE-CERQual group.

**Study status and estimated timeline**

At the time of submission, this review is in its preparatory phase. No formal data collection, screening, or data extraction has yet been conducted. Only preliminary searches have been performed across the selected databases to refine the search strategy and eligibility criteria.

The planned timeline for the review is as follows:

•Record screening: Expected to begin in March 2025 and to be completed by April 2026.

•Full-text review and data extraction: To commence in April 2026 and conclude by May 2026.

•Data synthesis and manuscript preparation: To take place between June and August 2026.

•Submission of the results manuscript: Anticipated for September 2026.

The timeline may be adjusted if necessary to maintain methodological rigor and ensure comprehensive coverage of the qualitative evidence base. All methodological procedures—including record screening, data extraction, and synthesis—will be conducted according to the protocol described in this document.

## Discussion

This qualitative meta-analysis seeks to address a critical gap in the literature by exploring how people living with HIV experience psychological interventions aimed at common mental health problems. While existing evidence highlights the effectiveness of various psychotherapeutic approaches, their acceptability from the perspective of participants—particularly those affected by self-stigma—remains poorly understood. By synthesizing first-person narratives, this review will provide valuable insights into what makes interventions feel appropriate, helpful, or challenging. The use of Timulak and Creaner's meaning-oriented synthesis allows for an in-depth understanding of participants' lived experiences across diverse contexts and populations. Findings will inform the design and delivery of more acceptable and person-centered psychological interventions. In doing so, this study may contribute to improving adherence, engagement, and long-term mental health outcomes in people living with HIV.

## Supporting information

**S1 Table. Full SPIDER-based search strategy for MEDLINE Complete (EBSCOhost).**
(PDF)

**S1 File. Prisma.**
(DOCX)

## Author contributions

**Conceptualization:** Jaime Garcia-Iglesias, Cristian Ortega, Alemka Tomicic.

**Investigation:** Cristian Ortega.

**Methodology:** Jaime Garcia-Iglesias, Cristian Ortega, Alemka Tomicic.

**Project administration:** Cristian Ortega.

**Supervision:** Jaime Garcia-Iglesias, Alemka Tomicic.

**Validation:** Cristian Ortega, Felipe Concha, Francisca Mena.

**Writing – original draft:** Cristian Ortega, Felipe Concha, Francisca Mena.

**Writing – review & editing:** Jaime Garcia-Iglesias, Cristian Ortega, Felipe Concha, Francisca Mena.

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
