## [Decision Letter · Decision Letter 0]

5 Feb 2026

Dear Dr.  Garcia-Iglesias,

Thank you for submitting your manuscript to PLOS ONE. After careful consideration, we feel that it has merit but does not fully meet PLOS ONE’s publication criteria as it currently stands. Therefore, we invite you to submit a revised version of the manuscript that addresses the points raised during the review process.

We look forward to receiving your revised manuscript.

Kind regards,

Satabdi Mitra, M.D(Community Medicine )

Academic Editor

PLOS One

Journal Requirements:

“This Qualitative Meta-Analysis is funded by University Diego Portales, Centre of Research in Clinical Psychology and Psychotherapy (CEPPS) and ANID – Millennium Science Initiative Program / Millennium Institute for Research on Depression and Personality - MIDAP AIM23_0002”

6. Please include a copy of Table 1 which you refer to in your text on page 8.

7. Please include captions for your Supporting Information files at the end of your manuscript, and update any in-text citations to match accordingly. Please see our Supporting Information guidelines for more information: http://journals.plos.org/plosone/s/supporting-information .

Reviewer's Responses to Questions

**Comments to the Author**

1. Does the manuscript provide a valid rationale for the proposed study, with clearly identified and justified research questions?

Reviewer #1: Yes

2. Is the protocol technically sound and planned in a manner that will lead to a meaningful outcome and allow testing the stated hypotheses?

Reviewer #1: Yes

3. Is the methodology feasible and described in sufficient detail to allow the work to be replicable?

Reviewer #1: Yes

4. Have the authors described where all data underlying the findings will be made available when the study is complete?

Reviewer #1: Yes

5. Is the manuscript presented in an intelligible fashion and written in standard English?

Reviewer #1: Yes

You may also provide optional suggestions and comments to authors that they might find helpful in planning their study.

Reviewer #1: This protocol addresses an important gap in HIV care by focusing on the subjective "acceptability" and lived experiences of participants in psychological interventions. The methodology is reasonably robust and well prepared.

However, there is a significant discrepancy regarding the inclusion period for studies. The abstract states the review will include studies published within the last 10 years , whereas the methods section specifies a limit of the last 25 years to reflect contemporary treatment contexts.

I am of opinion that the protocol is suitable for publication once this is addressed

**Do you want your identity to be public for this peer review?** For information about this choice, including consent withdrawal, please see our Privacy Policy

Reviewer #1: No

---

## [Author Response · Author response to Decision Letter 1]

23 Feb 2026

We sincerely thank you and the reviewer for the careful evaluation of our manuscript and for the positive and constructive feedback.

We are pleased that the reviewer considered the research question valid, the methodology technically sound, and the protocol sufficiently detailed and replicable.

Reviewer #1 noted a discrepancy between the time frame described in the abstract (10 years) and that described in the Methods section (25 years). We have carefully reviewed the manuscript and corrected this inconsistency. The abstract has been revised to specify that the review will include studies published between January 1, 2000 and February 23, 2026, consistent with the Methods section.

In addition, we have addressed all journal requirements outlined in the decision letter, including:

• Updating the Role of Funder statement to comply with PLOS ONE guidelines.

• Providing a complete Data Availability Statement in the submission form.

• Ensuring that the abstract in the manuscript and submission system are identical.

• Including the detailed MEDLINE (EBSCOhost) search strategy as S1 Table.

• Adding Supporting Information captions at the end of the manuscript.

• Reviewing and standardizing all references according to PLOS ONE formatting guidelines.

We believe that these revisions fully address the reviewer’s comment and the journal requirements. We thank you again for your consideration.

---

## [Editor Report · Decision Letter 1]

26 Feb 2026

Experiences of people living with HIV who have participated in psychological interventions: Protocol for a Qualitative Meta-Analysis

PONE-D-25-38173R1

Dear Dr. Jaime Garcia-Iglesias,

We’re pleased to inform you that your manuscript has been judged scientifically suitable for publication and will be formally accepted for publication once it meets all outstanding technical requirements.

Kind regards,

Satabdi Mitra, M.D(Community Medicine )

Academic Editor

PLOS One
---

## [Editor Report · Acceptance letter]

PONE-D-25-38173R1

PLOS One

Dear Dr. Garcia-Iglesias,

I'm pleased to inform you that your manuscript has been deemed suitable for publication in PLOS One. Congratulations! Your manuscript is now being handed over to our production team.

Kind regards,

on behalf of

Dr Satabdi Mitra

Academic Editor

PLOS One